# Polyharmonic Vibrations of Human Middle Ear Implanted by Means of Nonlinear Coupler

**DOI:** 10.3390/ma14185121

**Published:** 2021-09-07

**Authors:** Rafal Rusinek, Joanna Rekas, Katarzyna Wojtowicz, Robert Zablotni

**Affiliations:** Department of Applied Mechanics, Mechanical Engineering Faculty, Lublin University of Technology, 20-618 Lublin, Poland; j.rekas@pollub.pl (J.R.); katarzyna.wojtowicz@pollub.edu.pl (K.W.); r.zablotni@pollub.pl (R.Z.)

**Keywords:** middle ear implant, nonlinear coupler, ear dynamics

## Abstract

This paper presents a possibility of quasi-periodic and chaotic vibrations in the human middle ear stimulated by an implant, which is fixed to the incus by means of a nonlinear coupler. The coupler represents a classical element made of titanium and shape memory alloy. A five-degrees-of-freedom model of lumped masses is used to represent the implanted middle ear for both normal and pathological ears. The model is engaged to numerically find the influence of the nonlinear coupler on stapes and implant dynamics. As a result, regions of parameters regarding the quasi-periodic, polyharmonic and irregular motion are identified as new contributions in ear bio-mechanics. The nonlinear coupler causes irregular motion, which is undesired for the middle ear. However, the use of the stiff coupler also ensures regular vibrations of the stapes for higher frequencies. As a consequence, the utility of the nonlinear coupler is proven.

## 1. Introduction

Hearing loss is a very serious problem these days as reported in [1]. In the United States alone, approximately 30 million people are affected by hearing loss. More than one third of people over 65 years of age suffer from significant hearing deficits. Technological advances over the last decade have greatly improved the usability of conventional hearing aids that are not clearly defined in the literature. Usually, they are sound-amplifying devices designed to aid people who have a hearing impairment. These devices are non-invasive (not requiring surgery) and are placed behind the pinna, in the canal, or are body-worn. However, conventional hearing aids have weak points and disadvantages. Otolaryngologists and patients usually list acoustic feedback, occlusion effect, frequent battery changes, infection, discomfort, lifestyle restrictions, poor sound quality and even difficulty with speech recognition in crowded situations [1]. Moreover, conventional hearing aids can be used only in the case of mild to moderate hearing loss. Patients with deeper hearing loss (50 to 90 dB) have to find other, more technologically advanced devices, such as implantable middle ear hearing devices (IMEHDs), also known in short as middle ear implants (MEIs). Since IMEHDs can be used for both conductive and sensorineural hearing loss, they are becoming more and more popular [2].

Nowadays, there are several middle ear implant systems, which differ in terms of construction and mode of operation, including a fully implantable hearing prosthesis—the Otologics Middle Ear Transducer Carina^TM^ (Boulder, CO, USA), a semi-implantable device—the Vibrant Soundbridge (Med-El Corporation, Insbruk, Austria) [3], the Otologics Middle Ear Transducer (Boulder, CO, USA) [4]) and the Ototronix MAXUM system (Ototronix, Houston, TX, USA) [5,6]. Apart from their functionality (improved gain, sound quality, hearing and noise as well as acoustic feedback elimination), another important feature of these active prostheses is that they are made of a biocompatible material, i.e., titanium [7]. The materials from which implants are made should be appropriately selected biomaterials with high biocompatibility. Such materials should not cause acute or chronic reactions or inflammation. Furthermore, the materials for the implant should exhibit, among others, appropriate mechanical strength, fatigue strength, stiffness and abrasion resistance, and they should also be corrosion-resistant in the tissue environment. It is important that the biomaterial retains all its mechanical, physical and chemical properties during operation. Various materials have been used in the technique of hearing aid implantation over the years, including gold, stainless steel, tantalum, glass ceramics, alumina ceramics and hydroxyapatite [8]. Today, titanium and its alloys are often used in active middle ear implants. Titanium is an extremely light (specific weight 4.5 g/cm${}^{3}$) and stiff material. Its high mechanical stability and a simultaneous compatibility with bodies is due to the fact that a pure titanium surface immediately forms a thin layer of titanium oxide on contact with oxygen from air or water. This passive, ceramic layer protects the material, makes it immune to external influences and provides the necessary interface in biological tissue [9]. The Vibrant Soundbridge is the most useful and best-known prosthesis in the world. This partially implantable hearing device with electromagnetic transducer technology consists of an external part, a sound processor, and an implanted vibrating ossicular prosthesis (VORP) part [2]. The VORP, which is predominantly made of titanium, consists of a receiver, a conductive connector and a floating mass transducer (FMT). The FMT is usually attached to the long or short process of the incus by means of a coupler (clip), also made of titanium. Most studies on MEIs are experimental research studies with practical clinical conclusions. Different methods for attaching the FMT to the stapes head and footplate are applied [10,11,12]. For example, the problem of an optimal attachment point at the incus is discussed in [13]. The authors found that the incus coupler may be as good as that attached at the stapes. Schraven et al. [10] attached the FMT to the short process of the incus and compared this position with the standard attachment at the incus long process. The attachment of the FMT to the incus long process with the long process coupler results in generally good mechanical and functional coupling in temporal-bone preparations with a notable disadvantage between 1.8 and 6 kHz. Due to its elastic clip attachment, it is expected that the LP coupler can reduce the risk of necrosis of the incus long process. A review of the literature reveals the importance of the transducer attachment for sound transfer and a lack of an analytical approach to modeling and analyzing the problem of coupler stiffness. Therefore, this study focuses on the effect caused by nonlinear properties of the incus long process coupler and it is an extension of the authors’ previous study [14], which analyzed the implanted middle ear system with a linear coupler. The coupler nonlinearity may come from material or constructional properties, which is described in the next section for the coupler made of titanium (the commonly used material) and shape memory alloy (SMA). Nonlinear stiffness characteristics are used to ensure a more stable attachment of the coupler to the ossicular chain. However, given the fact that the nonlinear coupler may provoke unacceptable, nonlinear behavior patterns of the ossicles, this study analyzes dynamical effects induced by the nonlinear coupler. Taking all of the above, the novelties of this paper are:A new concept of a nonlinear coupler that has more stiff characteristics than the classical one;The first dynamic analysis of the middle ear with a nonlinear coupler;Finding regions of parameters where polyharmonic and chaotic stapes motion is possible.

This paper is organized as follows: Section 2 presents a five-degrees-of-freedom (5dof) model of the human middle ear with an implant affixed to the incus by means of a nonlinear titanium long process coupler. In Section 3, the effect of the coupler stiffness variation on both the middle ear and floating mass transducer vibration is described. The proposed model is analyzed for two cases of the normal and the pathological ear. Due to the key role of low frequency vibrations in speech recognition, the main (first) resonance is treated with special care. Finally, Section 4 offers a discussion of results and provides some final conclusions.

## 2. Nonlinear Ear Model

The human middle ear is composed of three ossicles, i.e., the malleus ($m_{M}$), the incus ($m_{I}$) and the stapes ($m_{S}$), as shown in Figure 1. The bones are connected to each other by means of joints: the incudo-mallear (IMJ) and the incudo-stapedial (ISJ); and to the temporal bone by ligaments: the anterior malleal (AML), posterior incudal (PIL) and stapedial annular (AL). Damping and stiffness properties of the joints and ligaments are denoted by $c_{*}$ and $k_{*}$, respectively. The middle ear is attached to the tympanic membrane whose viscoelastic properties are described by $k_{TM}$ and $c_{TM}$. On the opposite side, the middle ear is attached to the cochlea. The properties of the cochlea fluid are denoted by $k_{c}$ and $c_{c}$. The AL is assumed to have nonlinear stiffness characteristics, as reported in [15]. This middle ear model (without the implant) was also used in [16] to analyze the intact ear. Validation of the model based on the experimental results was successful, and therefore the model is used in this paper for modeling the implanted middle ear (IME). To obtain the IME model, the FMT (gray part in Figure 1) is attached to the incus by means of a coupler. Then, a five-degrees-of-freedom (5dof) model of the IME was obtained, as presented in Figure 1a. A similar model was also used in [17], where the IME was analyzed under different excitation conditions with a fixed coupler stiffness coefficient. The present study proposes a new concept of the coupler with constructional or material cubic nonlinearity presented schematically as the dashed line in Figure 1b. The former nonlinearity can be achieved using a typical (commonly used) titanium coupler of special design that provides increasing stiffness with a deflection. For instance, the stiffness of the coupler can be piece-linear (the solid line in Figure 1b). The latest one is represented, e.g., by shape memory alloy (SMA), which is used sometimes in a middle ear prosthesis [18] to fix a prosthesis head to the stapes arch. The SMA coupler could be a self-clamp element when heating without any external force. The SMA element used in the middle ear is described as a five-order polynomial, but the newest findings indicate that a three-order function is enough for small vibrations that occur in the ear structure [19,20].

Thus, the clip (coupler) is described by the linear ($k_{CLIP}$) and nonlinear ($k_{CLIP3}$) stiffness coefficients and the linear (viscous) damping $c_{CLIP}$ in Equation (Equation 2).

To build a more realistic FMT representation, the silicon rubber suspension of the magnet ($M_{m}$) in the can ($M_{c}$) is described by the third-order polynomial with the coefficients ${\tilde{\gamma }}_{45}$ and ${\tilde{\beta }}_{45}$. Silicon is assumed to be nonlinear according to the results reported in [21,22]. Thus, the governing differential equations of the presented system in the dimensional form are as follows:(1)$$\begin{array}{c} {\ddot{x}}_{M}m_{M}+{\tilde{k}}_{11}x_{M}+{\tilde{k}}_{12}x_{I}+{\tilde{c}}_{11}{\dot{x}}_{M}+{\tilde{c}}_{12}{\dot{x}}_{I}=0\\ {\ddot{x}}_{I}m_{I}+{\tilde{k}}_{21}x_{M}+{\tilde{k}}_{22}x_{I}+{\tilde{k}}_{23}x_{S}+{\tilde{k}}_{24}x_{c}+{\tilde{\gamma }}_{24}{\left(x_{I}-x_{c}\right)}^{3}+\\ {\tilde{c}}_{21}{\dot{x}}_{M}+{\tilde{c}}_{22}{\dot{x}}_{I}+{\tilde{c}}_{23}{\dot{x}}_{S}+{\tilde{c}}_{24}{\dot{x}}_{c}=0\\ {\ddot{x}}_{S}m_{S}+{\tilde{k}}_{32}x_{I}+{\tilde{k}}_{33}x_{S}+{\tilde{c}}_{32}{\dot{x}}_{I}+{\tilde{c}}_{33}{\dot{x}}_{S}+{\tilde{\gamma }}_{3}{x_{S}}^{3}=0\\ {\ddot{x}}_{c}m_{c}+{\tilde{k}}_{42}x_{I}+{\tilde{k}}_{44}x_{c}+{\tilde{k}}_{45}x_{m}+{\tilde{c}}_{42}{\dot{x}}_{I}+{\tilde{c}}_{44}{\dot{x}}_{c}+{\tilde{c}}_{45}{\dot{x}}_{m}-\\ {\tilde{\gamma }}_{24}{\left(x_{I}-x_{c}\right)}^{3}-{\tilde{\beta }}_{45}{\left(x_{c}-x_{m}\right)}^{2}+{\tilde{\gamma }}_{45}{\left(x_{c}-x_{m}\right)}^{3}=0\\ {\ddot{x}}_{m}m_{m}+{\tilde{k}}_{54}x_{c}+{\tilde{k}}_{55}x_{m}+{\tilde{c}}_{54}{\dot{x}}_{c}+{\tilde{c}}_{55}{\dot{x}}_{m}+{\tilde{\beta }}_{45}{\left(x_{c}-x_{m}\right)}^{2}-\\ {\tilde{\gamma }}_{45}{\left(x_{c}-x_{m}\right)}^{3}=P\cos\left(\omega t\right) \end{array}$$
where:(2)$$\begin{array}{c} {\tilde{k}}_{11}=k_{TM}+k_{AML}+k_{IMJ},{\tilde{k}}_{12}=-k_{IMJ},{\tilde{c}}_{11}=c_{TM}+c_{AML}+c_{IMJ},\\ {\tilde{c}}_{12}=-c_{IMJ},{\tilde{k}}_{21}=k_{12},{\tilde{k}}_{22}=k_{PIL}+k_{ISJ}+k_{IMJ}+k_{clip},{\tilde{k}}_{23}=-k_{ISJ},\\ {\tilde{k}}_{24}=-k_{clip},{\tilde{\gamma }}_{24}=k_{clip3},{\tilde{c}}_{21}=-c_{IMJ},{\tilde{c}}_{22}=c_{PIL}+c_{ISJ}+c_{IMJ}+c_{clip},\\ {\tilde{c}}_{23}=-c_{ISJ},{\tilde{c}}_{24}=-c_{clip},{\tilde{\gamma }}_{24}=k_{clip3},{\tilde{k}}_{32}=k_{23},\\ {\tilde{k}}_{33}=k_{AL}+k_{ISJ}+k_{C},{\tilde{c}}_{32}=c_{23},{\tilde{c}}_{33}=c_{AL}+c_{ISJ}+c_{C},{\tilde{\gamma }}_{3}=k_{AL3},\\ {\tilde{k}}_{42}=k_{24},{\tilde{k}}_{44}=k_{clip}+k_{m},{\tilde{k}}_{45}=-k_{m},{\tilde{c}}_{42}=c_{24},{\tilde{c}}_{44}=c_{clip}+c_{m},\\ {\tilde{c}}_{45}=-c_{m},{\tilde{\gamma }}_{45}=k_{m3},{\tilde{\beta }}_{45}=k_{m2},{\tilde{k}}_{54}=k_{45},\\ {\tilde{k}}_{55}=k_{m}=-k_{54},{\tilde{c}}_{54}=c_{45},{\tilde{c}}_{55}=c_{m}=-c_{54}. \end{array}$$
Now , the nonlinear coupler is defined by $\gamma_{24}{\left(x_{2}-x_{4}\right)}^{3}$ in Equation (Equation 1) (the second and fourth equations). Next, the dimensionless time τ, the frequency Ω and the coordinates $x_{1}-x_{5}$ are introduced according to the following expressions:(3)$$\begin{array}{c} \tau =\omega_{0}t,\omega_{0}=\sqrt{k_{AML}/m_{M}},\Omega =\omega /\omega_{0},\\ x_{1}=x_{M}/x_{st},x_{2}=x_{I}/x_{st},x_{3}=x_{S}/x_{st},x_{4}=x_{c}/x_{st},x_{5}=x_{m}/x_{st} \end{array}$$
Then, the dimensionless equations of motion take the form:(4)$$\begin{array}{c} {\ddot{x}}_{1}+k_{11}x_{1}+k_{12}x_{2}+c_{11}{\dot{x}}_{1}+c_{12}{\dot{x}}_{2}=0\\ {\ddot{x}}_{2}m_{2}+k_{21}x_{1}+k_{22}x_{2}+k_{23}x_{3}+k_{24}x_{4}+c_{21}{\dot{x}}_{1}+c_{22}{\dot{x}}_{2}+c_{23}{\dot{x}}_{3}+c_{24}{\dot{x}}_{4}+\\ \gamma_{24}{\left(x_{2}-x_{4}\right)}^{3}=0\\ {\ddot{x}}_{3}m_{3}+k_{32}x_{2}+k_{33}x_{3}+c_{32}{\dot{x}}_{2}+c_{33}{\dot{x}}_{3}+\gamma_{3}{x_{3}}^{3}=0\\ {\ddot{x}}_{4}m_{4}+k_{42}x_{2}+k_{44}x_{4}+k_{45}x_{5}+c_{42}{\dot{x}}_{2}+c_{44}{\dot{x}}_{4}+c_{45}{\dot{x}}_{5}\\ -\gamma_{24}{\left(x_{2}-x_{4}\right)}^{3}-\beta_{45}{\left(x_{4}-x_{5}\right)}^{2}+\gamma_{45}{\left(x_{4}-x_{5}\right)}^{3}=0\\ {\ddot{x}}_{5}m_{5}+k_{54}x_{4}+k_{55}x_{5}+c_{54}{\dot{x}}_{4}+c_{55}{\dot{x}}_{5}+\\ \beta_{45}{\left(x_{4}-x_{5}\right)}^{2}-\gamma_{45}{\left(x_{4}-x_{5}\right)}^{3}=p\cos\left(\Omega \tau \right) \end{array}$$
where the new dimensionless parameters are defined as follows: (5)$$\begin{array}{c} k_{11}={\tilde{k}}_{11}/\left(m_{M}\omega_{0}^{2}\right),k_{12}={\tilde{k}}_{12}/\left(m_{M}\omega_{0}^{2}\right),c_{11}={\tilde{c}}_{11}/\left(m_{M}\omega_{0}\right),\\ c_{12}={\tilde{c}}_{12}/\left(m_{M}\omega_{0}\right),k_{21}={\tilde{k}}_{21}/\left(m_{M}\omega_{0}^{2}\right),k_{22}={\tilde{k}}_{22}/\left(m_{M}\omega_{0}^{2}\right),\\ k_{23}={\tilde{k}}_{23}/\left(m_{M}\omega_{0}^{2}\right),k_{24}={\tilde{k}}_{24}/\left(m_{M}\omega_{0}^{2}\right),c_{21}={\tilde{c}}_{21}/\left(m_{M}\omega_{0}\right),\\ c_{22}={\tilde{c}}_{22}/\left(m_{M}\omega_{0}\right),c_{23}={\tilde{c}}_{23}/\left(m_{M}\omega_{0}\right),c_{24}={\tilde{c}}_{24}/\left(m_{M}\omega_{0}\right),\\ \gamma_{24}={\tilde{\gamma }}_{24}{x_{st}}^{2}/\left(m_{M}\omega_{0}^{2}\right),\\ k_{32}={\tilde{k}}_{32}/\left(m_{M}\omega_{0}^{2}\right),k_{33}={\tilde{k}}_{33}/\left(m_{M}\omega_{0}^{2}\right),c_{32}={\tilde{c}}_{32}/\left(m_{M}\omega_{0}\right),\\ c_{33}={\tilde{c}}_{33}/\left(m_{M}\omega_{0}\right),\gamma_{3}={\tilde{\gamma }}_{3}{x_{st}}^{2}/\left(m_{M}\omega_{0}^{2}\right),\\ k_{42}={\tilde{k}}_{42}/\left(m_{M}\omega_{0}^{2}\right),k_{44}={\tilde{k}}_{44}/\left(m_{M}\omega_{0}^{2}\right),k_{45}={\tilde{k}}_{45}/\left(m_{M}\omega_{0}^{2}\right),\\ c_{42}={\tilde{c}}_{42}/\left(m_{M}\omega_{0}\right),c_{44}={\tilde{c}}_{44}/\left(m_{M}\omega_{0}\right),c_{45}={\tilde{c}}_{45}/\left(m_{M}\omega_{0}\right),\\ \gamma_{45}={\tilde{\gamma }}_{45}{x_{st}}^{2}/\left(m_{M}\omega_{0}^{2}\right),\beta_{45}={\tilde{\beta }}_{45}x_{st}/\left(m_{M}\omega_{0}^{2}\right),\\ k_{54}={\tilde{k}}_{54}/\left(m_{M}\omega_{0}^{2}\right),\\ k_{55}{\tilde{k}}_{55}/\left(m_{M}\omega_{0}^{2}\right),c_{54}={\tilde{c}}_{54}/\left(m_{M}\omega_{0}\right),c_{55}={\tilde{c}}_{55}/\left(m_{M}\omega_{0}\right),\\ \omega =\Omega \omega_{0},q=Q/\left(m_{M}x_{st}\omega_{0}^{2}\right),p=P/\left(m_{M}x_{st}\omega_{0}^{2}\right). \end{array}$$
Since some parameters of the FMT were analyzed in [14,17], this study investigates only the problem of nonlinear clip stiffness for two variants of system damping, $c_{\cdots }$ and $c_{1\cdots }$. The first one is typical for the normal ear ($c_{\cdots }$), while the other, with decreased damping ($c_{1\cdots }$), in relation to normal ones, is typical for the pathological ear, e.g., one with incus luxation.

The problem of coupler design is of vital importance, as it allows proper FMT attachment. Manufacturers are constantly working on new coupler designs to improve fixation reliability and ensure better sound transmission. New design solutions may be expected in the near future. In light of the above, this study investigated the effect of cubic coupler stiffness as a scientific novelty in the field middle ear mechanics.

## 3. Polyharmonic Motion of the Implanted Ear

The dynamic behavior of the IME with the cubic stiffness coupler is examined near the primary resonance (Ω=1) for two cases: the normal and the pathological middle ear. Only an area of the first (primary) resonance is analyzed because of its important role in speech recognition. Results of numerical simulations are presented in the form of bifurcation diagrams in which points are collected at the zero velocity in order to additionally show the polyharmonic response of the system. Moreover, the classical phase diagrams with Poincar$\overset{´}{e}$ points are plotted for selected bifurcation parameters.

The middle ear and the FMT parameters used in the numerical simulations are given in Table 1. Three variants of external excitation were analyzed: *p*, 5p and 10p, where $p=1.5\times 10^{-4}$ ($P=1.2\times 10^{-4}$ N). A numerical model of the system was built using the MATLAB Simulink software package. The numerical simulations were performed by means of the Runge–Kutta fourth-order integration method (ode45) with a relative tolerance of $1\times 10^{-10}$ and a variable step size.

The nonlinear clip stiffness is characterized by $\gamma_{24}$, which is taken as a bifurcation parameter. The stapes vibration is analyzed primarily due to its importance regarding sound transfer to the inner ear. Moreover, the stapes motion is compared to that of the FMT elements.

### 3.1. Normal Ear

The normal middle ear with the implant (parameter $c_{\cdots }$ in Table 1) is not sensitive to the nonlinear stiffness $\gamma_{24}$. For all investigated excitations variants (p,5p,10p), the stapes motion is regular without extra harmonics, except for a small region near $\gamma_{24}=3\times 10^{6}$ at excitation 10p (Figure 2. Two lines in the figure indicate harmonic motion where the trajectory crosses zero velocity two times. The FMT vibrations are polyharmonic (Figure 3) despite the fact that the excitation is harmonic.

Regular attractors of the stapes motion are shown in Figure 4. Nevertheless, some symptoms of polyharmonic motion are observed in Figure 4b (the blue curve). A similar behavior pattern can be visible for the can and the magnet of the FMT in Figure 5, while an increase in $\gamma_{24}$ causes the phenomenon of attractor crossing for the excitation values of 10p and 5p (Figure 6). Interestingly, the period of vibration remains the same and equals 1T, which corresponds to the excitation frequency.

The polyharmonic oscillations shown in the bifurcation and phase diagrams near the first resonance also occur outside of the resonance. Therefore, the areas of polyharmonic motion are marked with gray color in Figure 7, Figure 8 and Figure 9. The stapes is free from polyharmonics when the system is excited by the force of 1p (Figure 7a). When the force is 5p, the polyharmonic vibrations occur near Ω = 2 (Figure 7b), whereas at 10p the gray regions can also be observed before the first resonance (Figure 7c). When the frequency is higher than Ω = 3 the stapes motion is fully harmonic.

Both the can and the magnet are free from polyharmonic vibration under an excitation of 1p (Figure 8a and Figure 9a). When the excitation is 5p, the can exhibits polyharmonic vibrations at Ω = 1 and Ω = 2 (Figure 8b), while the magnet shows this behavior only near Ω = 1 (Figure 9b). Under an excitation of 10p, the polyharmonic regions increase on the left of the plots toward the low excitation frequency (Figure 8c and Figure 9c).

### 3.2. Pathological Ear

A different situation is observed for the case of the pathological ear (parameter $c_{1\cdots }$ in Table 1), because quasi-periodic vibrations of the stapes occur at low $\gamma_{24}$ and 10p (Figure 10). The stapes motion is quasi-periodic and, in addition, polyharmonic (Figure 11a) at the same time. Thus, a stronger excitation causes a quasi-periodic motion of the stapes but only for a relatively small coupler nonlinearity. By increasing the nonlinearity to $\gamma_{24}=2\times 10^{6}$, the quasi-periodicity disappears and only a polyharmonic response is observed (Figure 11b).

It should be noted that neither the can nor the magnet of the FMT show the symptoms of quasi-periodicity for the same value of the $\gamma_{24}$ parameter when the stapes motion is quasi-periodic. Then, only the can shows symptoms of polyharmonic vibration in the trajectory Figure 12a and Figure 13a), while the magnet motion is devoid of this effect (Figure 12b and Figure 13b).

In the case of the pathological ear, the polyharmonic motion of the stapes (Figure 14a) and the can (Figure 15a) occurs even at a low value of excitation (1p), while the magnet is free from polyharmonics. The gray regions increase when increasing the excitation force (see Figure 14a–c, Figure 15a–c and Figure 16a–c). However, the magnet polyharmonic vibration region is smaller than that of the stapes and the can. When (Ω)>3, the gray regions disappear regardless of the nonlinearity described by $\gamma_{24}$.

The pathological ear exhibits a more complex behavior than the normal ear, including chaotic motion. Chaos may be hard to notice in the bifurcation diagrams because the diagrams were created for one set of initial conditions. For clarity, the system dynamics is therefore illustrated in a more comprehensive way using the maximal Lyapunov exponent (LE, Figure 17). Chaos occurs when $\gamma_{24}<1.2$ and when $\gamma_{24}>1.8$. Interestingly, a higher excitation force can sometimes lead to regular motion, whereas a higher nonlinearity causes motion irregularity, although this may not be clear when looking at the bifurcation diagram shown in (Figure 10). It is expected, however, that a change in the excitation amplitude and frequency will also lead to chaotic motion. The regions of irregular motion are shown in Figure 18 as a two-parameter plot. The black areas indicate Ω and *p* where the maximal value of the Lyapunov exponent is positive. Thus, chaos exists near Ω = 1, 2, 3, 4 and 7, whereas at Ω>7 the system is free from chaos, whatever the excitation amplitude (*p*). The Ω range between 7 and 10 can be regarded as unconditionally regular.

## 4. Summary and Conclusions

This paper described a new titanium coupler with nonlinear, piece-linear characteristics that was modeled by the cubic nonlinearity. It has been found that both the coupler stiffness and the excitation force have a key meaning for polyharmonic motion of the stapes and parts of the FMT as well. However, the theoretical analysis of the implanted middle ear dynamics has revealed a more complex behavior, especially for the case of the pathological ear with decreased damping properties. Nonlinear stiffness even results in chaotic motion of the ossicular chain in the range of low frequencies (below 3 kHz). The area of polyharmonic vibration decreases when the excitation force is lower. In contrast to the pathological ear, the behavior pattern of the normal middle ear is much more predictable, because of regular vibrations, especially for small excitation values that are typical for a normal human life. The effect of the coupler’s nonlinearity on the middle ear dynamics is less significant than that of the excitation force and frequency.

Summing up, the coupler with nonlinear characteristic can be used in medical practice because the risk of unpredictable stapes motion may only occur in pathologically changed ears with decreased damping, or may be due to very high excitation, which, however, is not commonplace in real life. From a scientific point of view, the study provides some interesting conclusions:Periodic excitation in the nonlinear multi-degrees-of-freedom system causes harmonic, polyharmonic and even chaotic vibrations;Small damping is a cause for irregular motion;Polyharmonic response depends on the excitation amplitude and frequency of excitation and the nonlinearity level. The stronger the nonlinearity, the more irregular the motion;Generally, strong excitation in nonlinear systems causes chaotic vibrations, but here, a smaller amplitude of excitation produces sometimes irregular motion, whereas a strong one makes the system response regular.

Finally, it must be stressed that the problem of nonlinear magnet suspension and electromechanical coupling between the transducer and the middle ear has been omitted in the proposed model, and should therefore be thoroughly analyzed in the future. 

## Figures and Tables

**Figure 1 materials-14-05121-f001:**
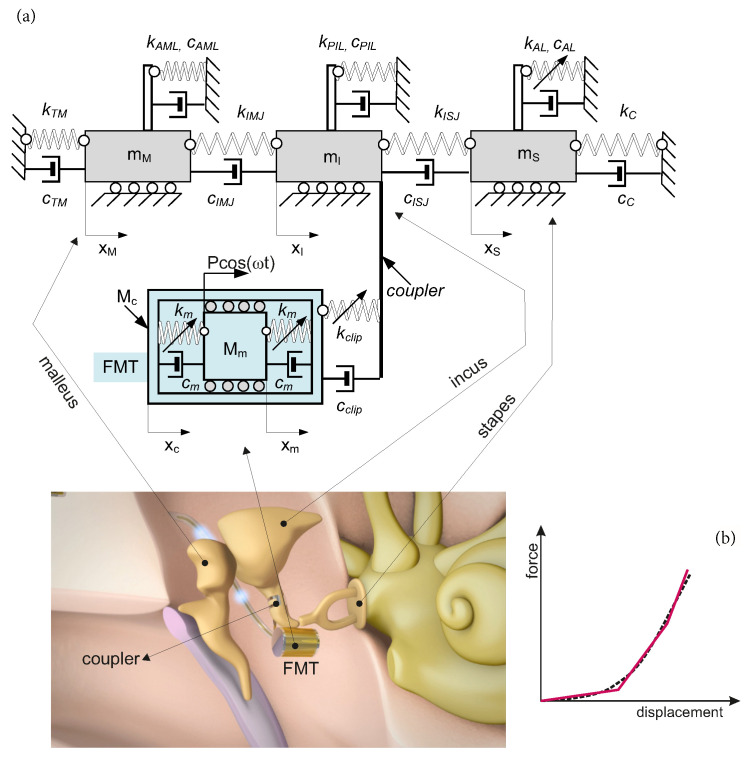
Five-degrees-of-freedom model of the IME (**a**), and the nonlinear characteristic of the coupler (**b**).

**Figure 2 materials-14-05121-f002:**
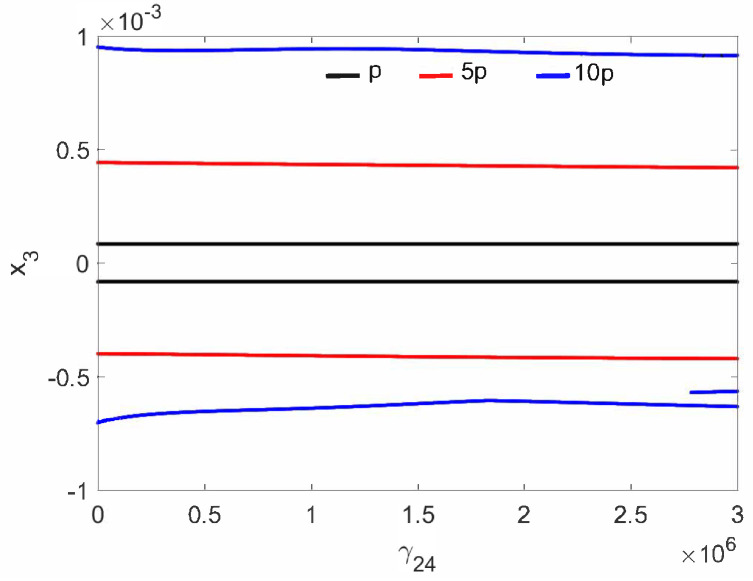
Bifurcation diagrams of the stapes in the normal ear at the first resonance versus the nonlinear stiffness of the coupler ($\gamma_{24}$).

**Figure 3 materials-14-05121-f003:**
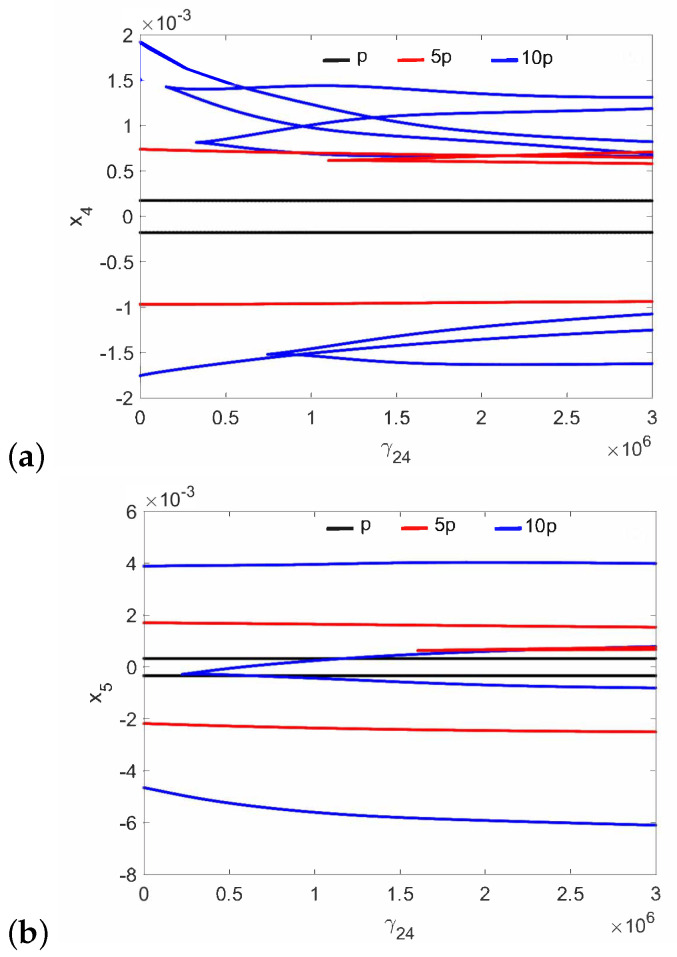
Bifurcation diagrams of the FMT in the normal ear at the first resonance versus the nonlinear stiffness of the coupler ($\gamma_{24}$): the can (**a**), and the magnet (**b**).

**Figure 4 materials-14-05121-f004:**
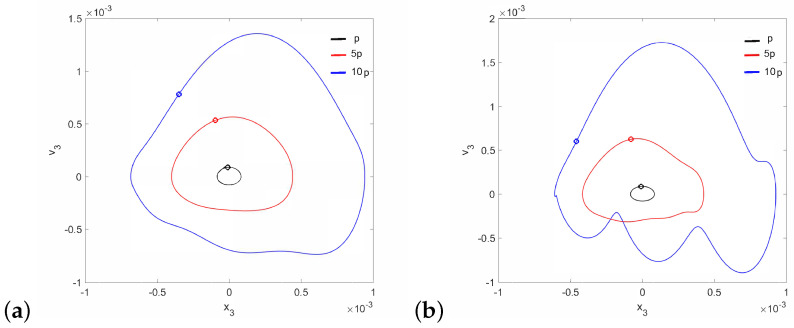
Phase diagrams of the stapes motion near the first resonance in the normal ear for $\gamma_{24}=1\times 10^{5}$ (**a**) and $\gamma_{24}=2\times 10^{6}$ (**b**).

**Figure 5 materials-14-05121-f005:**
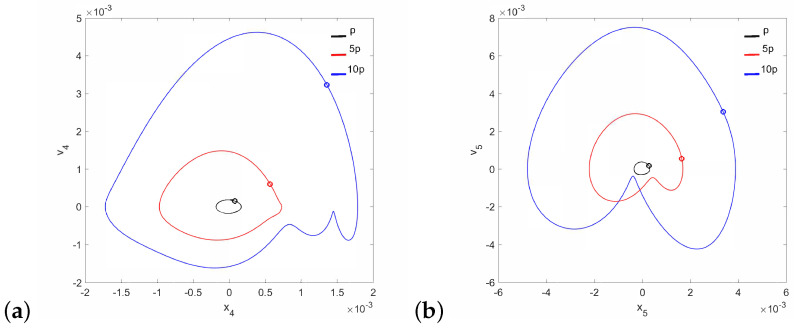
Phase diagrams of FMT motion near the first resonance in the normal ear for $\gamma_{24}=1\times 10^{5}$: the can (**a**) and the magnet (**b**).

**Figure 6 materials-14-05121-f006:**
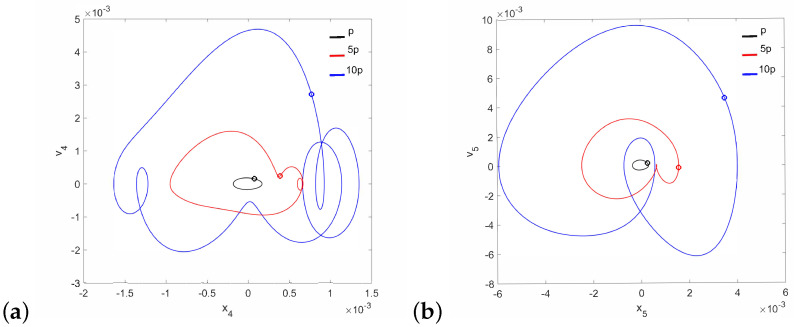
Phase diagrams of FMT motion near the first resonance in the normal ear for $\gamma_{24}=2\times 10^{6}$: the can (**a**) and the magnet (**b**).

**Figure 7 materials-14-05121-f007:**
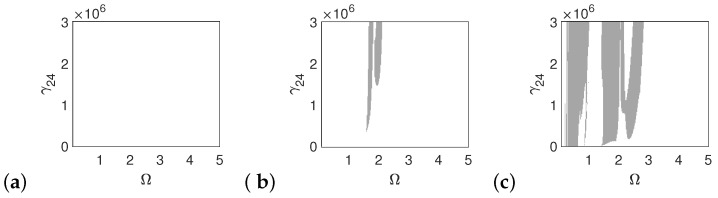
Regions of polyharmonic vibrations of the stapes for the case of the normal ear versus the nonlinear stiffness rate of the coupler ($\gamma_{24}$) and the excitation frequency (Ω) under the excitations *p* (**a**), 5*p* (**b**) and 10*p* (**c**).

**Figure 8 materials-14-05121-f008:**
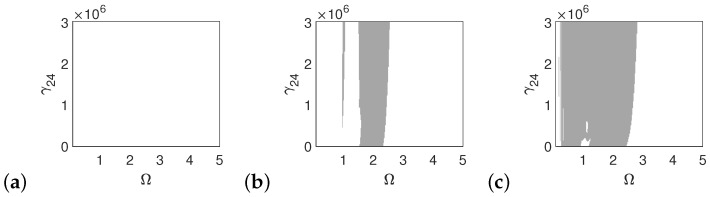
Regions of polyharmonic vibrations of the can for the case of the normal ear versus the nonlinear stiffness rate of the coupler ($\gamma_{24}$) and the excitation frequency (Ω) under the excitations *p* (**a**), 5*p* (**b**) and 10*p* (**c**).

**Figure 9 materials-14-05121-f009:**
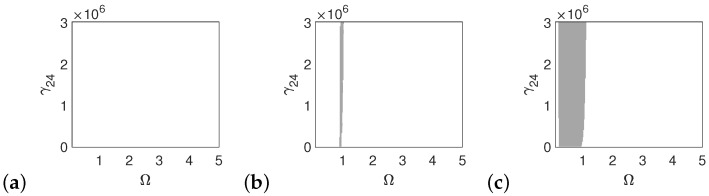
Regions of polyharmonic vibrations of the magnet for the case of the normal ear versus the nonlinear stiffness rate of the coupler ($\gamma_{24}$) and the excitation frequency (Ω) under the excitations *p* (**a**), 5*p* (**b**) and 10*p* (**c**).

**Figure 10 materials-14-05121-f010:**
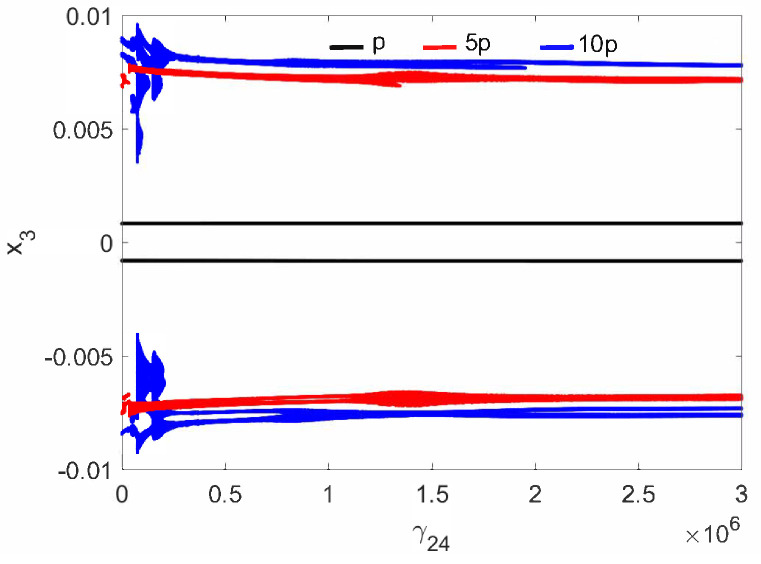
Bifurcation diagrams of the stapes motion in the pathological ear versus the nonlinear stiffness rate of the coupler ($\gamma_{24}$).

**Figure 11 materials-14-05121-f011:**
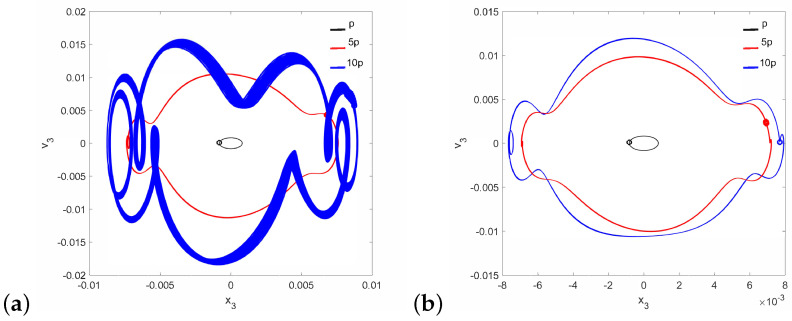
Phase diagrams of the stapes motion in the pathological ear for $\gamma_{24}=1\times 10^{5}$ (**a**) and $\gamma_{24}=2\times 10^{6}$ (**b**).

**Figure 12 materials-14-05121-f012:**
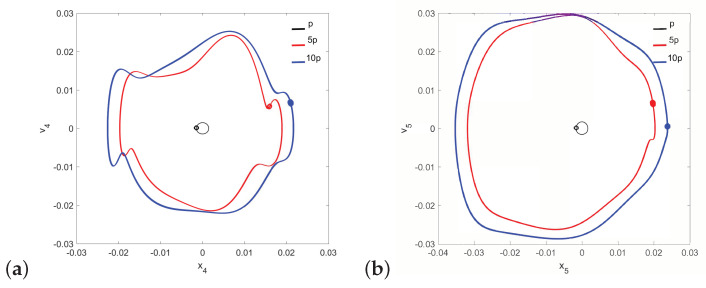
Phase diagrams of FMT motion in the pathological ear for $\gamma_{24}=1\times 10^{5}$: the can (**a**) and the magnet (**b**).

**Figure 13 materials-14-05121-f013:**
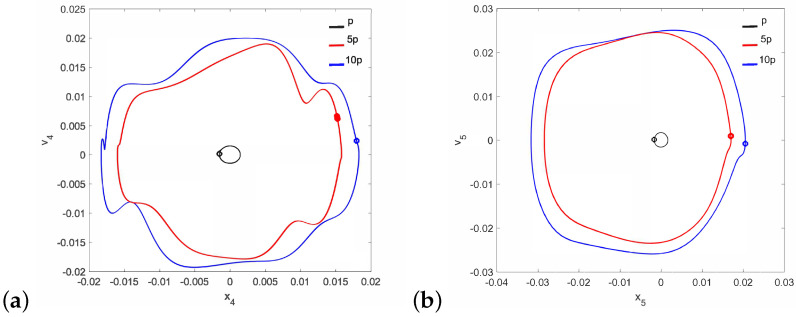
Phase diagrams of FMT motion in the pathological ear for $\gamma_{24}=2\times 10^{6}$: the can (**a**) and the magnet (**b**).

**Figure 14 materials-14-05121-f014:**
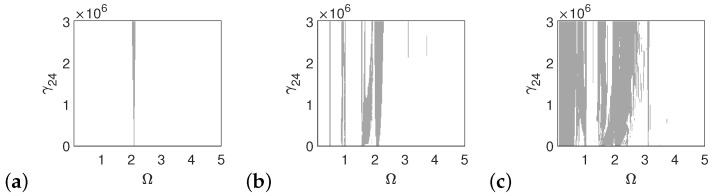
Regions of polyharmonic vibrations of the stapes in case of the pathological ear versus the nonlinear stiffness rate of the coupler ($\gamma_{24}$) and the excitation frequency (Ω) under excitations of *p* (**a**), 5*p* (**b**) and 10*p* (**c**).

**Figure 15 materials-14-05121-f015:**
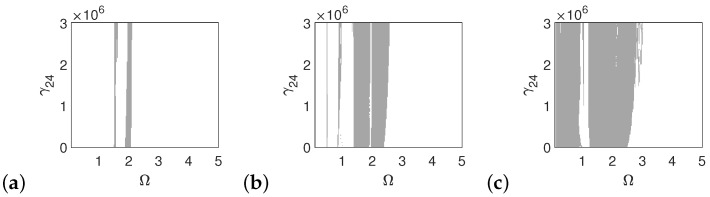
Regions of polyharmonic vibrations of the can in case of the pathological ear versus the nonlinear stiffness rate of the coupler ($\gamma_{24}$) and the excitation frequency (Ω) under excitations of *p* (**a**), 5*p* (**b**) and 10*p* (**c**).

**Figure 16 materials-14-05121-f016:**
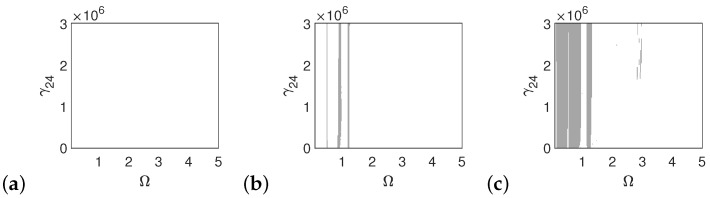
Regions of polyharmonic vibrations of the magnet in case of the pathological ear versus the nonlinear stiffness rate of the coupler ($\gamma_{24}$) and the excitation frequency (Ω) under excitations of *p* (**a**), 5*p* (**b**) and 10*p* (**c**).

**Figure 17 materials-14-05121-f017:**
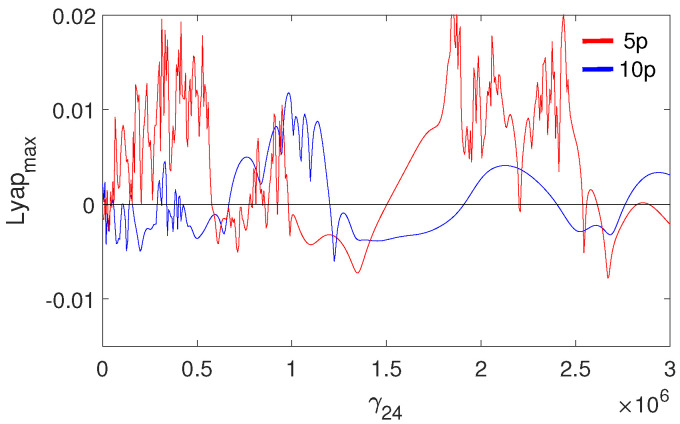
Maximal Lyapunov exponent for the pathological ear versus the nonlinear stiffness of the coupler ($\gamma_{24}$).

**Figure 18 materials-14-05121-f018:**
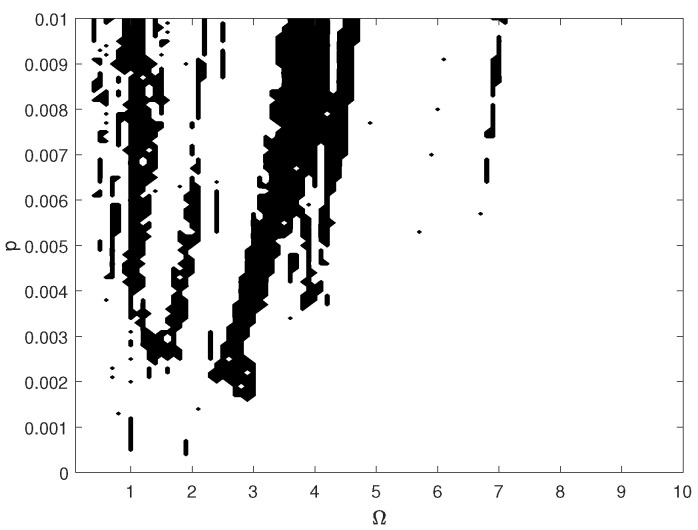
Map of the positive maximal Lyapunov exponent for the pathological ear versus excitation frequency (Ω) and amplitude (*p*).

**Table 1 materials-14-05121-t001:** Parameters of the middle ear model with the FMT taken from [14,17].

Mass *m* [mg]	*k* [mN/μm]	*c* [mNs/mm] (Normal)	$k_{*3}$ [Ns${}^{3}$/mm${}^{3}$] $k_{*2}$ [Ns${}^{2}$/mm${}^{2}$]	$c_{1}$ [Ns/mm] (Pathological)
$m_{M}=25$	$k_{TM}=0.3$	$c_{TM}=60$		$c_{1TM}=0.359$
$m_{I}=28$	$k_{AML}=0.8$	$c_{AML}=125$		$c_{1AML}=0.538$
$m_{S}=1.78$	$k_{IMJ}=1000$	$c_{IMJ}=359$		$c_{1IMJ}=28.86$
$M_{c}=5$	$k_{PIL}=0.4$	$c_{PIL}=55$		$c_{1PIL}=0.981$
$M_{m}=5$	$k_{ISJ}=1.35$	$c_{ISJ}=7.9$		$c_{1ISJ}=0.039$
	$k_{AL}=0.623$	$c_{AL}=0.04$	$k_{AL3}=0.013$	$c_{1AL}=0.033$
	$k_{C}=0.2$	$c_{C}=1.7$	$k_{m2}=0.188$	
	$k_{m}=0.85$	$c_{m}=5$	$k_{m3}=0.014$	
	$k_{clip}=2.0$	$c_{clip}=10$	$k_{clip3}=2.25$	

## Data Availability

The data presented in this study are available on request from the corresponding author.

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
