# Peer review of "Polyharmonic Vibrations of Human Middle Ear Implanted by Means of Nonlinear Coupler"

_materials, 2021, doi:10.3390/ma14185121_

Round 1

Reviewer 1 Report

REVIEW

of the manuscript

Polyharmonic Vibrations of Human Middle Ear Implanted by Means of Titanium Coupler

By Rafal Rusinek, Joanna Rekas, KatarzynaWojtowicz and Robert Zablotni

This is an interesting paper developing a nonlinear model of the middle ear with a floating mass transducer as part of the implantable hearing device.

On the base of the developed 5 degree of freedom nonlinear model, the authors have studied the dynamics of system for a normal ear and for the pathological ear. They have proved arising of poyharmonic vibrations when the loading is harmonic, quasi-periodic and chaotic vibrations. These phenomena arise for different values of the excitation frequencies, the amplitude of the load and the values of the nonlinear clip stiffness characteristics γ24.

I have the following minor remarks:

1.In my copy of the manuscript, Figs. 17 and 18 are not visible. This does not allow me to confirm the authors’ conclusions about the chaotic motion.

  1. It will be better to define the dimensionless frequency Ω in eqs (3) where other dimensionless parameters are introduced.

  1. I recommend a check of the English. For example in some places are used “of” instead of “for”, “received” instead of “obtained” (page 3), etc.

I recommend the publication of the manuscript after improving it according these suggestions.

Author Response

Thank You very much for your review. Our answers are in the file.

Reviewer 2 Report

Report on the manuscript

Title:  Polyharmonic Vibrations of Human Middle Ear
Implanted by Means of Titanium Coupler
Rafal Rusinek, Joanna Rekas, Katarzyna Wojtowicz and Robert Zablotni

Manuscript ID: materials-1274115

I think the readers of this journal will appreciate the results of this manuscript.  Generally speaking, the manuscript is well written, the material is judiciously divided and organized and correct from scientific point of view. Some changes are, however, necessary. For these reasons I can recommend the acceptance of this paper after some corrections.

Before that the Editor makes a decision, I suggest that the authors emphasize take into account the following corrections

  1. The Abstract section is presented too superficially and without pointing out what is really being done. Please improve this section.
  2. Please highlight how the work advances or increments the field from the present state of knowledge and provide a clear justification for your work.
  3. The section Conclusions will be point out the original results of the paper and can be extended to highlight the contributions. Please provide a clear justification for your work in this section, and indicate uses and extensions if appropriate.
  4. The conclusion section has to be rewritten doing an effort to remark the main findings rather than summarizing the article content.
  5. Please check the paper again for any possible misprints.
  6. The text needs to be checked and revised by a native speaker or a language expert. You may consider (at your own cost) the use of a possible professional copyediting service
  7. After each relationship a point, comma or semi-column should be placed.
  8. I think the authors need to emphasize more clearly the contribution of the manuscript from a scientific point of view.
  9. Template of the journal must be respected.

If the author takes into account these observations the work can be published.

Author Response

(The authors gave the same response as above.)

Reviewer 3 Report

The authors are presenting a work on the nonlinear middle ear model using floating mass transducers as part of the implantable hearing device. The authors are considering a 5 dof model for the simulation and analysis of this device.

I believe the current study is of high importance and investigation is quite accurate in terms of mathematical background and scientific research.

However, I don't believe the present work deals well with the scope of materials since the present work discussed titanium application but no properties not experiments are shown in the manuscript since a discrete model is present without any detail of the real device.

Therefore I suggest to submit the present manuscript in a more suitable journal.

Author Response

(The authors gave the same response as above.)

Reviewer 4 Report

This paper study by nonlinear vibration of human middle ear implant by  Titanium coupler. The dynamic performance of normal and pathological ears is discussed by various parameters of the nonlinear model. The reader might be interesting on this subject, model, and results. However, some drawbacks have to be revised before acceptance in publication.

  1. The abbreviation is not clearly defined in manuscript, e.g. MET, MAXUM MED-EL, LP, and so on.
  2. The last paragraph on Section I. Introduction is not necessary existence.
  3. The correspondent ear model should be provided to illustrate on the KCM (spring-damper-mass) model in Fig. 1.
  4. As shown in Fig. 1(b) and description in Section 2, the nonlinear vibration is considered by various stiffness and viscoelastic material. The property in model and equation has to be clearer illustrated.
  5. How and why to define the normal and pathological ears should be clearer defined in Section 3.
  6. Figs. 17 and Figs. 18 are lost. It might be caused by that file size of all figure in manuscript is too large to transfer successfully. It suggested to re-paste all figures by selection of paste from "picture enhanced metafile" in "Paste Special".

Author Response

(The authors gave the same response as above.)

Round 2

Reviewer 3 Report

According to the comments and justifications given by the authors the manuscript should be accepted in the present form.